# "Who'd Have Thought?": Unravelling Ancestors' Hidden Histories and Their Impact on Dharug Ngurra Presences, Places and People

Jo Anne Rey 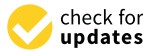

Department of Indigenous Studies, Faculty of Arts, Macquarie University, Wallumatta Campus, North Ryde, Sydney, NSW 2113, Australia; jo.rey@mq.edu.au

**Abstract:** As a means of opening the lid on transgenerational silencing—which was a survival strategy for thousands of Indigenous families against intended cultural genocide—while balancing the place of auto/biography in that journey, this paper focuses on the impact of Ancestors' hidden histories and how the discovery of those histories drives complex identifications when woven with Presences, places, and people on Dharug Ngurra/Country. Using my own family's recently uncovered early colonial Ancestral storying, histories that involve Dharug traditional custodian, African slave, and Anglo characters, some as First Fleet arrivals, the paper considers the place of auto/biography as a form of agency that brings past into presence, and which, in turn, opens opportunities to heal, decolonise, and transform Dharug and, more broadly, Indigenous communities, their knowledges, practices, and ontologies. When this activation involves most of the metropolis known as Sydney, Australia, we recognise its transformative potential to change non-Indigenous people's perspectives. When we recognise auto/biography as a form of 'truth-telling', it allows a space to re-story relationality, both human and other-than-human, and restores Indigenous presence into Ngurra for biodiverse justice in a climate-changing world. Addressing these matters through poetic multimedia allows a place of safety between the pain and the healing.

**Keywords:** Indigenous; activism; genealogy; identity; Dharug; storying; 'kin-sensing'

## 1. Introduction

This paper comes through Dharug Ancestral and Elder voices, past, present, and future. Dharug is the name given to the Aboriginal people who are the traditional custodians of Dharug Ngurra, which is the Dharug language term for Dharug Country: the lands, earth, waters, and sky, including all the associated Presences (physical and metaphysical) that form agency within the area that today comprises most of what is called the Sydney basin, Australia. With more than five million humans inhabiting the area today, caring for Country-as-city requires looking beyond the surface landscapes, narratives, and extinction industries. It requires seeing the continuities—the threads of connections—that have woven cultural pasts into surviving agency for sustainable futures. It is to be noted that 'Ancestors' and variations of the term, as well as other terms such as 'Elders' and 'Presences' are capitalised to show respect, and is also indicative of their metaphysical agency within cultural practices. By acknowledging the presence of Ancestral and Elder knowledges within this paper, the reader is challenged to look closely into the detail of the storied fabric (the Presences as well as the Absences) that have been woven across the continent of so-called 'Australia' today. As society walks towards a First People's Voice proposed by the existing government, it asks for a deeper sense of relationality. One that can only be achieved through an ethical praxis that has underpinned 65,000 years of continuing cultural care.

This paper focuses on the impact of Ancestors' hidden histories and how the discovery of those histories drives complex identifications when woven with Presences, places, and

people on Dharug Ngurra/Country. As such, it opens the lid on transgenerational silencing, which was a survival strategy for thousands of Indigenous families against intended cultural genocide. In the process, it aims to acknowledge the place of both autobiography and biography, expressed here as auto/biography. Using my own family's recently uncovered early colonial Ancestral legacies, involving Dharug traditional custodian, African slave, and Anglo-Romani heritages (some as First Fleet arrivals), the paper considers the place of auto/biography as a form of agency. One that brings past into presence, and which, in turn, opens opportunities to heal, decolonise, and transform Dharug and, more broadly, Indigenous communities, their knowledges, practices, and ontologies. When this activation involves Dharug Ngurra (my place of closest connection and belonging), we recognise its transformative potential to change non-Indigenous people's perspectives. When we recognise auto/biography as a form of 'truth-telling', it allows a space to re-story relationality, both human and other-than-human, while restoring Indigenous presence into Ngurra. This is important, I argue, given that we are in a context of climate-challenging futures. Restoring Indigenous relationalities supports biodiverse justice. 'Biodiverse justice' refers to and recognises the injustices done to the other-than-human species, which have been perpetrated across this continent since settler colonialism arose following the arrival of the British in 1788 (Beresford 2021; Griffiths 2018; Pascoe 2014; Tatz 2017). With climate-changing catastrophes already happening and likely to worsen within the next three decades, the urgency of restorative human–other-than-human relationality, as 'biodiverse justice', needs to be highlighted.

This paper addresses these matters using a poetic multimedia production, *Dharug Presences in Place: Where's Dolly Dreaming?* (Rey and Parry 2022). I suggest that artistic productions allow a safe space for non-Indigenous people to meet the pain suffered by Indigenous peoples and other-than-humans across the last 235 years on this continent. This meeting then opens the healing and reconciliation journey that is required for sustainable futures. Using the metaphor of 'Black Dolly' (a black rag doll) in the audio/visual poetic narration acts as the agent for one thread of my own Ancestral storying (Rey and Parry 2022). It is one that weaves together 19th-century childhood traumas with contemporary degradations of Ngurra created by colonising mentalities. It also resonates with ongoing State and corporate practices that perpetuate those traumas. This paper works through some of the complexities that truth-telling confronts, when for many descendants, Aboriginal and early colonial heritages are intertwined. It is argued that uncovering the legacies through hidden stories brings deeper connections to Presences, places, and people and not only assists in truth-telling but, through storying, opens up a path towards localised, sustainable relationality and activism.

The structure of this paper commences with the broadest contextual frame, which places auto/biography in the colonial his-storying by outlining its agency to disrupt the patriarchy in the process through personalised oral reflection. From that opening point of disruption, the focus turns to the inclusive place of other-than-humans—the Presences (metaphysical and physical) within Indigenous cultural praxis—introducing 'kin-sensing' to express relationality as within and beyond dominating narratives of 'them and us'. The lens then narrows further, to the specific poetic multimedia production that interweaves human auto/biography with particular Presences, and places, reflecting the Ancestral heritages of the author, while simultaneously entwining past, present, and futures. It concludes with the recognition that it is through the detail that we can sense the universal, and that it is in localised interweaving that those personalised Presences in place can be met and that decolonising activism resides, so that the grand patriarchal narratives can be unspun.

## 2. Context: Re-Cognising the Place of Auto/Biography in Colonial Storying

Childhoods are not always easy, no matter when you live. Even Prince Harry, Duke of Sussex has had his moments (Duke of Sussex Prince Harry 2023). However, for the first generation of British children, born during the voyage from England, or soon after the First Fleet's arrival in 1788 into Dharug Ngurra, the conditions must have been physically as

well as emotionally challenging to say the least. Some of those first children were offspring of the nearly 1500 human beings on those 11 ships (Holden 2000). Others were children of female convicts, born through early partnerships and Christian marriages. As Holden (2000, p. 138) notes, the 48 children (aged between a few days and 14 years) who arrived in 1788 made up 3.3 per cent of the (Anglo) new-arrival population, but by the end of the 18th century that number had jumped to 20 per cent. However, those statistics do not include the Dharug (and/or Dharawal) communities and their children already living here. Given that traditional custodians had previously been living in their clan areas of connection, caring, and belonging with their families, inside their multilinguistic culture, secure in their place in the world, their sense of childhood must have been exceptionally peaceful and free in comparison (Rey 2021)—that is, at least until the smallpox epidemic of 1789, when an estimated 50 to 90 per cent of the local population was wiped out (Deacon 2020). Others, like my four-times great-grandmother, Frances Randall (c.1793), were born from partnerships between First Fleet men and Dharug women. Unions such as these were not registered in any written Christian marriage record-keeping systems. Instead, they are recorded within Dharug community transgenerational oral storying.

In an analysis of 18th-century childhoods and the conditions that dominated them, Robert Holden draws on the conditions in London and the context of child chimneysweeps to illustrate one example of how poverty-stricken children survived and ended up being criminalised by the legal system of the day. Some did end up on the First Fleet, being transported to the other side of the world and arrived into a completely unrecognisable context on 26 January 1788 (Holden 2000). Holden contemplates how the inevitable meeting between a British child transportee and an Aboriginal child would have impacted both. What did that cultural gulf feel and look like?

Understanding those cross-cultural contexts through children's eyes draws us into our own experiences of difference before we were told how to respond. That is not to render all children's experiences as those of innocent witnesses and/or victims. As Faulkner (2016) notes, the depiction of children across historical, legal, and interpretative narratives is more a place of conflation and ambiguity than clarity and innocence. Given that the rate of illiteracy in Britain (England and Wales) in 1800 for males was around 40 per cent, and for females, 60 per cent, the reliability (or lack thereof) of the written record, has to be brought into question. Demanding written records as the source for identification for this period ignores the reality of the historical context (Lloyd 2007). Such ignorance (some would argue determined ignorance), along with other continuing acts of colonisation, has had a disastrous impact on Dharug and neighbouring families and children. It is often only recently, through family storying behind closed doors and through genealogical databases which identify descendants, that the silencing can be stopped, and truth-telling can surface. As the national political landscape today (at time of writing) calls for an Indigenous Voice to Parliament, the question of truth-telling and of facing the hidden stories and the auto/biographies becomes currently relevant and continuously critical for national healing.

One autobiographical uncovering of such hidden stories surfaced relatively recently, through the historical documentation of the First Fleet Black African convicts, which included the historian's version of the life of my five-times great-grandfather John Randall senior (Pybus 2006). Randall's story was one of six Black African First Fleet convicts, which had previously been unknown to the public. My Ancestor was raised in Stonington, Connecticut, fought for the British in the War of Independence, and then, when that was lost, was transported to England (perhaps to escape ongoing slavery), only to end up being transported as a convict to Australia. I argue that this is one example of previously silenced diversity within the colonising population which strengthened the white colonial 'Australia' narrative. It follows that relying on 'evidence-based' written records for identity verification has real-life consequences, when people suddenly uncover the hidden oral stories of their Ancestors, now that it is considered 'safe' to do so. Yet, so often, when people seek to identify as Indigenous as a means of connection, caring, and developing a

sense of communal belonging, they find themselves rejected by the Western system that privileges the written-only form of evidence-based 'truth'.

The insistence on the sourcing of 'truth' using this narrow interpretation continues the trauma associated with what in Australia is known as the 'Stolen Generations'—those practices that took children away from their families, culture, lands, and languages (Commonwealth of Australia 1997). This perpetuates the suffering associated with colonising domination. Given that most convicts were illiterate (in written forms, as noted above) and that only certain elites within the British population could write and read, it can be argued that when it comes to reliance on early colonial-settler identifications, transgenerational oral biographical information is no less reliable than the written records that are available, because their scarcity makes the gaps, the missing information, louder than the 'facts'. While written records may help *if*, and *when,* they are available, their absence should not be considered as the legitimate measure for identity verification when it comes to the earliest colonial-settler invasion of Dharug and neighbouring Dharawal and Gundungurra countries. The organisation Link Up, for example, created to bring removed Aboriginal people and their missing families together, admits that they are an 'evidence-based' body and, because of this, can only have limited success in achieving their goal (Personal Email. Received: 3 February 2023).

Rather, I argue that providing a space for oral history should be permissible. Yet, choosing how to best tell those stories can be a major work, one that involves unravelling the colonial and continuing mythscape woven across society by the State education systems. I contend that it is only in the specific place-related storying, those auto/biographies through place, that intimate knowledges of connection can shine through. This paper contends that by focusing on the domestic scenarios, the storying of childhoods, mothers, and grandmothers, on the lived realities dwelling behind closed doors in particular places, that we can begin to see beyond the patriarchal myths and grand empiric narratives.

## 3. Disrupting Grand Narratives and Patriarchal 'Truth-Telling'

Colonial cultural norms dominated the mentalities of those arriving on the shores of Dharug Ngurra (and those of her coastal neighbours) in 1788 and well beyond the entire transportation era, which ceased in 1868. Some would say that those colonising mentalities continue today. The vast bulk of written histories in the scholarship, up until the 1970s, represented the foundational empiric white patriarchal values. However, as Lake (2013, p. 190) notes:

> When a new wave of women's history burst onto the Australian national scene in the 1970s, its angry tone, revolutionary critique, and national political focus, reflected its close connections with the women's liberation movement.

Works such as Summers' (2002) *Damned Whores and God's Police* (originally published in 1975) challenged male dominance in the field of perpetrating colonising His-story, by focusing on 'Her-storying'. Yet, for most of the population, and inter-generationally, it would be the grand masculine mythmaking of the 'heroes' of colonisation which would infect the education systems, established predominantly through the second half of the 19th century, and then continued and reinforced across 'White Australia' in the first half of the 20th century. It should also be noted that until the 1970s it was legal for Aboriginal children to be excluded from schooling (Commonwealth of Australia 1997), and so any inclusion of Aboriginal perspectives in the curriculum did not commence until at least the 1980s.

However, change in perspectives can be seen through those tracking the historical records and accounts (Brook and Kohen 1991; Brook 1999; Djoric 2011; Docker 2015; Karskens 2010, 2020; Reynolds 1987; Tobin 1999), the geographic knowledges (ACF 2020; Ashton 2000; Currie and Willoughby City Council 2008), core Indigenous ways of knowing, being, and doing (Moreton-Robinson 2015; Watson et al. 2014; Watson 2015), and many other sources. This tracking shows a continuing thread that entwines the Presences (as more-than-humans) across times, the places across geographies, and the people across

events. Doing so renders a sense of continuity, connection, caring, and belonging that can culturally support and sustain Aboriginal children and their childhoods—even, and especially, when so many have been removed from their families, from their first language, and from their places of belonging.

As an example, auto/biographical accounts in the Australian government's 'Bringing Them Home' report relate personal recollections by Aboriginal adults of the trauma they experienced as children, when they were taken from their families at a very early age (Commonwealth of Australia 1997). It becomes very clear in this report (Commonwealth of Australia 1997) that losing a sense of belonging to people, places, and storying drives many to seek reconnections to their kin. I argue that 'kin-sensing' can be a part of that journey when the circumstances prevent written evidence from being found. It is important to note at the outset that 'kin-sensing' does not aim to replace or represent Aboriginal kin systems which involve moiety, totem, and skin understandings of kin as foundational Aboriginal law/lore. It is a system that underpins cultural framing and social organisation (Watarrka 2023). Instead, 'kin-sensing', as I define it, is the intuitive connection we feel and recognise within and across Presences, places, and people. It aligns the sensory with the cognitive: a 'third place' of relationality, a place of poetic and artistic response (Rey and Harrison 2018).

## 4. Grounding Relationality through Dharug Ngurra

### 4.1. Kin-Sensing Relationality: A Poetic Production

At the human-to-human level, 'kin-sensing' is the process of hearing others' storying and recognising the resonances with your own. At the human–other-than-human level, it is sensing the agency of the place, and the Presences within that place. It is by being attuned sensorily to the sights, sounds, texture, smells, calmness, and tensions of the creatures and species in that place, and by being consciously aware of their impacts and agency upon you, that a relationality arises. Sometimes, that sensory experience creates its own storying and narrative, intimating its own auto/biography and affectively inscribing you in the process. Just as poetic language resonates, and can impact us, and affects us in an embodied way, 'kin-sensing' can also prompt a poetic embodied response. Thus, my use of 'kin-sensing' focuses on sensing relationality, sensing connection, in whatever moment or context. I will discuss this further below, but in finding and recognising our kin—whether that be through written genealogies to human ancestors, intuitively through attentive presence with other-than-humans (Plumwood 1993; Rose 2013), and/or experiential hybrid perception (Kim et al. 2015)—that recognition opens us to negotiate the traumas of our own and others' auto/biographies as we seek to relate and to contextualise our lives with those being met.

Within Indigenous context and speaking of the human-to-human realm, the performance, as well as the form, of the auto/biography is an important component of Indigenous resistance. As Cariou (2016, p. 314) describes it:

> *"Life-telling"* . . . *[as the] uniquely oral aspects of Indigenous oral traditions and everyday storytelling practices* . . . *is crucial in the struggle to resist the colonization of Indigenous knowledge.*

Entwining the relationalities that underpin life-telling enables sustainability and fosters wellbeing (human and other-than-human) through a sense of community. As Cariou (2016) further notes:

> *For an Indigenous person to tell one's story is to affirm, against the genocidal history of colonialism, "We are still here." To listen to such a story is to understand, in an embodied and active way, that Indigenous life-telling is fundamentally a medium of relationships, one that binds people together and affirms their connections to the land.*

Choosing that form, or combinations of forms, becomes an act of agency. While many writers have focused on settler colonialism, with its genocidal intentions and outcomes delineated (Moreton-Robinson 2000; Reynolds 1987; Tatz 2017; Wolfe 2006), as a form of important truth-telling, the written form is one that favours the Western model of objectified

and historical knowledge construction and representation as 'truth'. Privileging this model ignores the personalised performative practice of truth-telling, and instead, as Cariou (2016) rightly points out, diminishes Indigenous orality, positioning it as indicative of vulnerability and also as unverifiable, thereby becoming inadequate, in the eyes of English law, for 'truth-telling'. Yet, when politicians visit communities and, in places such as Uluru and at the Garma Festival, hear and see the 'truth-telling', they respond with the recognition of the validity and truth of those Presences in place (Australia Government 2022).

### 4.2. Weaving Dharug Contextual Continuities

Dharug life-telling, as auto/biography, has been presented in several key literary works. Grace Karskens (2020) entwined Dharug oral biographies with historical place-based written knowledges, both archival and geographical, to articulate a deep sensory knowledge of and connection with the Dyarrubin (Hawkesbury/Nepean) river system among the Dharug community. Rey's (2019) doctoral thesis focused on seven Dharug women's own storying, as a collection of autobiographical place-related perspectives, one of which was the author's own. Together, the women wove their relationality with their places of significance, creating an oral storied Dharug women's landscape that was heard, and then parts of which were represented in written form. The outcome was to demonstrate the continuity of Dharug Presences, Dharug places, and Dharug practices across seven different locations within Ngurra/Country, while drawing out Dharug women's identification journeys in the process (Rey 2019).

Weaving those autobiographical perspectives impacted my own journey towards understanding my own place, the truths of my Ancestors, and their connections to their places across time. One example involved hearing from several Dharug women that they or their parents were informed that their background was 'Spanish', in order to explain the darker colouring of hair and skin (Rey 2019). This resonated, because as a child I was told I had 'French' ancestry. Any narrative that would cover for the "touch of the tar brush", as my father used to say of those with a possible Black heritage, would be employed in order to protect family members from the State authorities and other racist shaming. Finding those commonalities offered a point of connection, while simultaneously stimulating further questions. A sense of 'kin' and relationality arose.

However, that is not to say that everything in the Ancestral storying was, or is, known. Uncovering Ancestral pathways, and how they interweave with our Presences, is always an ongoing re-cognition: a contextual attentive focus. How we tell those stories today is part of both understanding our own strengths and limitations, and weaving the opportunities that are presented to us today.

To that extent, the opportunity to weave poetry, videography, and orality through the physical metaphor of 'Black Dolly' across today's landscapes became the next transformational chapter in my own recognition of personal relationality with Presences, places, and people. This process, I argue, strengthens identities and agency, and offers opportunities to transform conceptualisations of 'truth-telling'. It also opens pathways to 'kin-sensing'.

### 4.3. Where's Dolly Dreaming? A Poetic Multimedia Response (Rey and Parry 2022)

In mid-2022, the opportunity arose to create a cultural piece that would reflect continuing connections for an upcoming Indigenous cultural exhibition, "Songspirals", at Newcastle University Gallery (http://www.theuniversitygallerynewcastle.com.au/2022-uni-gallery.html, accessed on 28 September 2022). The cultural piece created for the exhibition, "*Dharug Presences in Place: Where's Dolly Dreaming?*", drew on support from a variety of sources that, together, created a poetic, audio/visual 'journey' that not only entwined Presences and places on Dharug Ngurra as city today, but also, through the metaphor of 'Black Dolly', wove the colonisation storying of those Presences, places, and people of the past, using one particular auto/biographical experience from my Dharug colonial ancestor, Ann Randall (Rey and Parry 2022). The poetic multimedia production became a journey that articulated my connectivity—the 'kin-sensing' across presence, place, and Ancestral

people. To that extent it represents the 'auto' thread in the term auto/biography. The collaboration with videographer Mark Parry and his role in producing the visual and audio aspects became the 'biography' thread. Together, they were interwoven to produce a poetic audio/visual spiral across time, space, and matter: a timespacemattering (Barad 2010).

## 5. Interweavings

### 5.1. People Threads

It is known through the written records that Ann Randall was placed in the Parramatta Female Orphan School in Parramatta in 1822, as a 6-year-old, by her mother Fanny Randall (Brook and Kohen 1991). She was later transferred to the Blacktown Native Institution (BNI) as a 9-year-old in 1825. She was one of the first seven children to be placed at the BNI as it was the second phase of Governor Lachlan Macquarie's drive to 'civilise' the Dharug and other Indigenous children after the failure of the first attempt, the Parramatta Native Institution (PNI) (Brook and Kohen 1991; Norman-Hill 2019). That storying resonates and intertwines with the continuing removal of Indigenous children from their homes and families today (Grandmothers Against Removal 2016).

Ann Randall was born in c.1816, the daughter of Fanny (Frances) Randall, who was the daughter of Black American First Fleet convict John Randall. John Randall originated from the slave-owning Randall family of Stonington, New Haven, Connecticut, and was born in 1764 (Pybus 2006). His daughter Frances Randall was born in 1792. While there are no written records of his partnership with Dharug Kitty, community oral storying confirming this has been passed down the generations through a variety of descendant families. Given that there were so few written records at the time, as discussed above, the likelihood of this being true, I argue, is equally, if not more, valid than not.

As Pybus (2006) and genealogical and archival records show, the Black convicts integrated in a variety of ways, particularly as between the Randall, Martin, and Aiken families, who intermarried. We know, for example, that Frances (Fanny in the diminutive form) Randall initially partnered with John Aiken, in c.1810, though we also know that that marriage had already ended when Aiken placed a notice in the Sydney Gazette in 1824, notifying that he would no longer be responsible for her debts as she had 'absconded'.

In the Sydney Gazette of 1824, Frances's husband, John Aiken, placed the following notice:

*'Whereas my wife, Frances Aiken, had absconded from her home without any cause or provocation, I do hereby caution all persons from harbouring, or giving credit to the said Frances Aiken on my account, as I will not be responsible for debts contracted by her. Parramatta. John Aiken 26 April 1824'*

(https://australianroyalty.net.au/tree/purnellmccord.ged/individual/I58373/Frances-Randall, accessed on 7 June 2023).

Given that Fanny had attempted to put both Ann and her older sister Eliza into the Female Orphan's School at Parramatta in 1822, there is a strong indication that she was already trying to extricate herself from the existing domestic situation a couple of years beforehand. While Ann was accepted, Eliza was refused, as she was already considered too old (Brook and Kohen 1991).

At some point, Fanny was to partner with white convict William Brown. It is conjectured that both Ann and Eliza may have been fathered by him, prior to her departure from Aiken. In any case, she was to settle with William Brown, and they had many offspring, one of whom was John Brown, who became a timber-getter within the Turramurra/Wahroonga forests around the source of what is today called the Lane Cove River. In Dharug language this is known as the Durrumburra/Turrumburra river. At one crossing point of the river, today known as Brown's Waterhole, it is recognised through physical evidence to be a Dharug significant zone and as such was a meeting place between Turrumburra and Wallumattagal people's clan areas. Its contemporary naming relates to John Brown who used to water his bullocks there as they dragged the felled timbers down to the salt water area

of the river, where they would be floated to the harbour and used for either trade or the construction of the Sydney infrastructure (Hawkins 1994).

In the meantime, we know that Ann (Randall-Aiken-Brown) survived her institutionalisation at the BNI and then, at some point, went on to live in the Windsor–Wilberforce area along the Dyarubbin (Hawkesbury River) area at the foot of the Blue Mountains, having many descendants today. While I am descendant from her first known partnership with George Smith-Holmes, others descend from her second partnership with George William Rose, while living at nearby Currency Creek, which was a reserve for Aboriginal families of the area until 1895 (History of Aboriginal Sydney 1895) (https://historyofaboriginalsydney.edu.au/timeline/west?keys=1895, accessed on 7 June 2023). Others continue from her third and final partnership with Charles Young, with whom she was to remain, at Wilberforce, to the end of her life (Rey 2019).

These details are important as the rationale underpinning the places of connection that were interwoven through the 'Black Dolly' narrative. It was my Ancestor Ann Randall's connections to the BNI and the Dyarubbin River, along with her mother Fanny Randall-Aiken-Brown's connections to Brown's Waterhole, that became the three places underpinning and weaving together 'Black Dolly's' oral and visual narrative. Together, along with the Presences in those places, the cultural work became an early outcome of my current postdoctoral research project, 'Weaving Country across the City: Activating Dharug Ngurra'.

Rather than use the term 'site' for each of these three places, I will use the term 'zone', as a way to decolonise the geographical and territorial language that diminishes the recognition of relationality which underpins our consciousness of the complexity of Ngurra (Andrew and Hibberd 2022).

### 5.2. Placing the Threads in Three Forms

The three places of significance for the postdoctoral research project (and those for the poetic/multimedia work) were selected because of their connective threads. Those connections, while also being personal in terms of my Ancestral storying, as discussed above, were sourced for their agency, as zones of activation.

### 5.2.1. Shaw's Creek Aboriginal Place

This place, located within the Yellomundee Regional Park, holds profound Dharug Ancestral storying and colonial his-story, as well as being a zone of current activism for the regeneration of Dharug culture for many decades now (https://www.nationalparks.nsw.gov.au/visit-a-park/parks/yellomundee-regional-park/learn-more#91FFE025A683425388B4A6F04AEA42D0, accessed on 7 June 2023).

The zone is located on what was colonial-settler Shaw's Farm, beside Shaw's Creek and where it meets the Dyarubbin/Nepean River. The zone was formally gazetted in 2014 as an Aboriginal Place due to the proximate engravings made by our Pleistocene people and ongoing community connection and activism to protect it and care for Country (Attenbrow 2010; Karskens 2020). However, it was also a zone of manufacture and massacre. As Low notes, (https://www.simplyaustralia.net/massacre-at-shaws-creek/, accessed on 7 June 2023) the "thousands of spears" sighted by the pack of settlers responsible for killing the local Aboriginal people were not just intended for killing settlers, as the settlers presumed, and as was reported in the Sydney Gazette at the time, but were being created for cultural hunting and fishing due to the proximity of resources, e.g., timber, stones, and gum sap used as a glue. Given the proximity of the site to fresh water from the river and the creek, there were ample life-sustaining resources, and it was/is this proximity to known Presences (other-than-humans) that enables cultural continuity.

As one Dharug research interviewee (211223) said of their relationship with this place:

> . . . so you get familiar, you become connected personally to Country, to Ngurra. So there's this, like for me, it's like this energy or this vibration happening. And it's, it's

*activating you, igniting you, in every way. And so, you, you see all these different levels. And, you know, well, . . . you want to see deeper, and* [into] *Ngurra.*

The activation is told through the example of undertaking a cultural burn:

*. . . So this is a really good example of coming and building this relationship up in a particular area where we've come and put our energy, and that is through the cultural burning, it's through where, you know, like, taking the weeds out, caring for her, watching her grow through the microlaena* [microlaena stipoides or Weeping Grass] *which is this beautiful native grass, seeing the birds come in, and being nurtured through feeding, seeing their babies come in, and then being nurtured in this space. So, it's this ongoing cycle of nurturing in here and looking after... And do you feel that in your body physically? Yes.* (Interview: 23 December 2021)

Having these contemporary experiences provokes a sense of autonomy and agency that speaks back to the domination of continuing colonising legacies that cross Dharug Ngurra in the form of urbanisation, state bureaucratic control, and multinational neo-liberal media narratives. Contemporary activation of Dharug Ngurra, through places of Ancestral presence and storying, therefore, offers opportunities for reweaving the threads of cultural continuity back into the city. As a form of truth-telling, it offers a new way to break down the barriers, between First Nations people and broader communities, for sustainable futures. It expresses personalised, localised activism.

5.2.2. Blacktown Native Institution (BNI)

The second zone chosen for its significant contemporary cultural activism is the BNI (DSMG 2020). Its agency continues beyond its colonial legacy of being a constructed institution for the removal and 'civilisation' of Dharug and other Indigenous children. Rather, it is recognised for its liveliness, as a source entwining humans and other-than-humans—a "Living, Embodied Being" (Andrew and Hibberd 2022). While extensive historical accounts of the BNI can be located (Brook and Kohen 1991; Norman-Hill 2019), it is only in recent years that contemporary Dharug custodial perspectives on the living significance of the place have been heard. Since 2013, those perspectives have been expressed through artistic events, informal gatherings, and, when the BNI was finally returned to Dharug community in 2018, full ceremony and a corroboree (DSMG 2020; GHD 2018; MCA 2023). As participant in the 2020 Biennale of Sydney, the BNI zone is recognised as a zone of variegated activeness, one in which:

*the exceptional approaches that Darug and other Aboriginal and Torres Strait Islander peoples have enacted to re-scape this place of Stolen Generation's memory and zone of trauma. . . . Darug community cultural practices are enacting and empowering a crucial healing process and laying the foundation for thriving futures.* (Andrew and Hibberd 2022, p. 169)

From Dharug custodians' perspectives:

*Nura (Country) speaks. The Blacktown Native Institution site is the artist.*

*Guided by her, as a site of Dreaming, her life, her ceremony and songlines. She represents identity, traumas, traditions.* (Andrew and Hibberd 2022, p. 171)

As such, the physical place holds and tells stories, from the past, to the present, to the future, and demonstrates the concept of a living Indigenous Be-Ing (Rey 2021)—one that enacts sustainable relationality, which always was and always will be agentic. Understanding place through the complexity of stories—the zones of diversity—that are held and heard, tangible and intangible, recognises its relationality with the humans who have sat within its embrace for thousands of years. At the same time, understanding this living relationality recognises the ecological and geological movements and changes that are interwoven within and across those Presences, places, and people. Transgenerationally, passing on those stories enables community to heal by moving beyond memorialisation of child removal and incarceration towards sustainable and resilient futures. Together, such

an interweaving moves us beyond presence, place, and people as objective separations to our opening up through artistic, shared connectivity. Through this, 'kin-sensing' comes to life: a woven timespacemattering (Barad 2010).

### 5.2.3. Brown's Waterhole (BWH)

The third zone of activism is the BWH. Before the family connection through Fanny/ Frances Randall's marriage to William Brown and their descendant, John Brown's, connections were known, the BWH called to be included in the postdoctoral research through a variety of ways. These included its proximity to Macquarie University, the presence of Ancestral engravings, the raucous cacophony of birdlife, and the Presence of its principal resident, "Edgy Eddy", the Eastern Water Dragon. The place of waterholes in Aboriginal law/lore was also a very strong point of connectivity.

However, beyond all these positives was the visible and visceral sense of sadness that drew me. Visible as rubbish blocking the Terry's Creek tributary. Sad as frothing filth forming at the junction with the Durrumburra (today known as the Lane Cove River) and the massive mounds of weeds colonising the banks. It was the human lack of caring that spoke loudest. No one appeared to understand the importance of this zone. Especially not the cyclists tearing down into the river valley in both directions, whizzing over the concrete concourse from both directions, east and west, screaming at people (stopped to admire the life of the site) to get out of their way, as they gathered break-neck speed before ascending the other side. Altogether, the BWH zone spoke viscerally, and before I knew anything specific about the pasts, human and other-than-human, or the futures, and the possibility of undertaking a cultural healing burn, it was the Presences—the good, the bad, and the ugly—that pulled me towards a sense of connection and belonging and a sense of active determination to care for Ngurra/Country, to care for 'kin' (Rey 2023).

Today, some five years after determining the need to care for Ngurra through the BWH zone, much attention, energy, and learning is unfolding, not least as preparation for the Dharug women's-led cultural burn, funded by the NSW Department of Planning and Environment's (DPE) Cultural Fire Management Team (CFMT) and undertaken through my postdoctoral research project. Such a journey has brought Dharug community into relationship with various local and government departments, including local shire councils, National Parks and Wildlife Services (NPWS), Macquarie University (MQU), and broader community allies. BWH as a Dharug place is activated as a meeting place through our traditional custodial lens: a first since colonisation. It has also enabled contemporary research equipment purchases, such as cameras and software, to determine what forms the local other-than-human residents take: the fauna, the flora, and the waterhole life. The training of Dharug women, provided by Macquarie University scientists, in the use of this equipment helps and prepares for future community research projects. Permissions and funding for camping by Dharug women and allies near the site have also established new layers of relationality. NPWS and RFS have been constant journey companions. The prospect of enacting a cultural fire drives the agency for these new connections, for new caring, and for strengthened belonging to Ngurra across the sites. It articulates the 'Ing' of activism, within the broader context of contemporary urbanity (Rey 2021).

### 5.3. Threading the Presences in Place

Life's agency has also been described as the 'Ing' that underpins identities (Rey 2021). Sensing 'kin' takes Plumwood's (1993) 'attentive presence' one step further by recognising a relationality between the enactor and the recipient of that attentive presence. Sensing 'kin' with Presences opens us to that 'Ing' or relational agency. Over time, holding that relationality with Presences informs what Rose (2013, p. 93), reviewing Val Plumwood's work, described as 'philosophical animism'. I propose here that it is through weaving the three zones (BNI, Shaw's Creek, and the BWH) with the Ancestral life narrative of 'Black Dolly' that a journey of 'philosophical animism' was undertaken, one which expresses a deeply Indigenous agency across time, place, and matter.

Firstly, there is Black Dolly's own Presence, as the central character in the multimedia piece. The role of Black dolls, beyond being childhood playthings, crosses cultures and continents, tingling with resonances of slavery, spirituality, and racism. As Santo (2019, p. 270) notes, "Objects are fundamental components of cosmology in Afro-Cuban religions; they serve to represent, pay homage to, and feed a constellation of covetous spirits." She argues that things are continuous with unfolding selfhoods, 'with, and as, *affects*' (Santo 2019, p. 272). Dolls become selves and extensions of the owner's self. As such, their agency resides in their affect, on their context, including humans. Presences in place impact and affect humans consciously or unconsciously, so that self becomes "constituted in webs of spirits and objects . . . an affective and historical network of person, spirits, and things, none of which can be easily disentangled" (Santo 2019).

Afro-Cuban religious dimensions become relevant to the Rey and Parry (2022) auto/biographical production as a Presence when we remember that Ann Randall's grandfather, John Randall Senior, was of African-American slave-family tradition. It also becomes particularly relevant when we recognise that a heritage of collecting Black dolls was active within the childhoods of Ann Randall's descendants. See Figure 1 below.

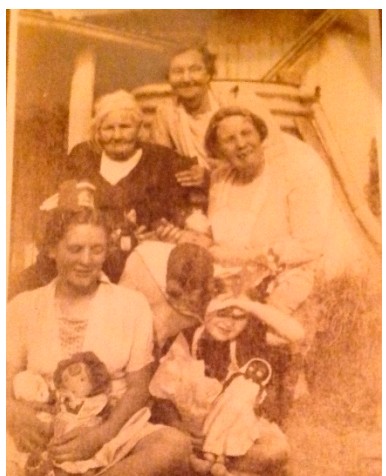

**Figure 1.** Ann Randall-Young descendants with black dolls (Rey 2019, p. 281). Image: C. Hunt.

Secondly, Presence can be seen through the interweaving of time, place, and matter at Shaw's Creek Aboriginal Place where the hand-ochre images inscribe the tree trunk (Rey and Parry 2022, 05:34). While they represent current cultural activism in place, their Presence resonates with Ancestral practice and as such carries an agency that can affect the viewer now, inscribing their consciousness into the future. As the visual text and poetics entwine, through verbal speed and intonation, as well as the choice of vocabulary, the viewer is inscribed more deeply, and so a connection to the presence, the place, and the person (of Ann Randall as Black Dolly) is strengthened. There are many examples of this in the piece: flooded river, broken trees, and Dolly's location amidst the twisted corrugated iron scraps, and 'embraced' in the broken branches of her hiding spot. A dramatic narrative Presence grows that consciously, or unconsciously, weaves the viewer into relationship with the production, and Dolly's journey.

These are just some of the other-than-human Presences that build the story, but significantly, I would argue that it is the contrasts as Presences of modernity (traffic congestion, barbed wire, torn plastic refuse, and a concrete tunnel as just some examples) that affect the viewer and cause them to face the realities of a tormented future, where security becomes only a possibility, a moment in time, and not anything that can last forever. As the final scene shows Dolly being carried into the tunnel (that holds a security camera on the roof), the text makes the temporary nature of the concept more real: "She found the way into the future that, *for now*, seemed safe". This weaves the viewer into a consciousness of the fragility that underpins current contextual critical realities, such as climate-changing

catastrophes. Activating philosophical animism, as a caring for Country through alternative ways of knowing, being, and doing, becomes an important option. I contend that it requires Indigenous ethical praxis.

## 6. Drawing Together the Threads: Current Contextual Criticalities

Indigenous cultural practice always has been and always will be centred on ethical relationality between humans and other-than-humans. The fact that human-produced climate change has only been a Presence across the last 250 years (approximately the same time as colonisation of the Australian continent) means the search for sustainable futures is critical (ACF 2020; Burke et al. 2016; IPCC 2021, 2023; Lyons et al. 2020).

Sustainable ethical relationality underpins caring for Country and is evidenced in the fact that Indigenous peoples of this continent have continued for more than 65,000 years. I call this the 'web of continuity', as expressed in Figure 2 below.

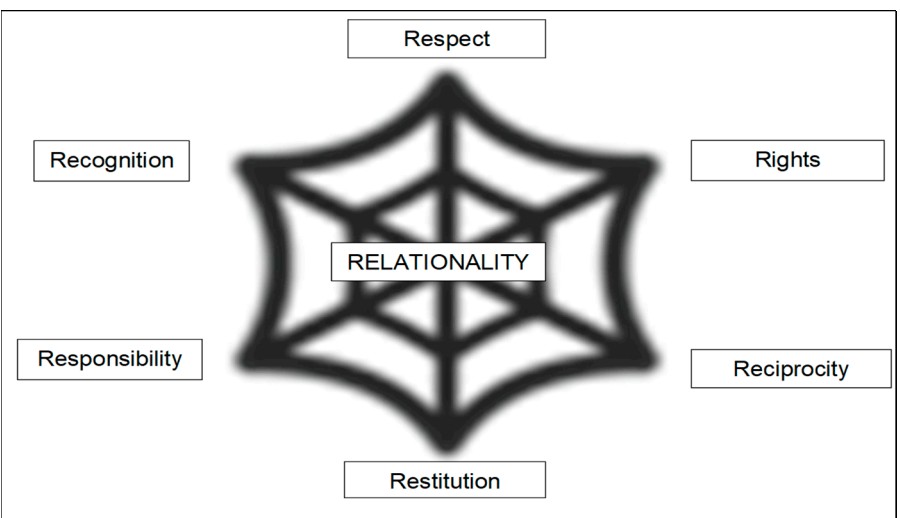

**Figure 2.** Ngurra as Sustainable Ethical Praxis: A Web of Relational Continuity. (Image: Rey 2023).

In the context of auto/biography as a form of 'truth-telling', I argue that recognising the relationality between humans and other-than-humans is critical to achieving sustainable futures across the sentient spectrum. I would further argue that this web of relationality has been driven underground by colonising perpetrators, whose disconnections have fostered unethical human-centricity (Wolfe 2006). I will now explain how this web is interwoven and why its activation is so important.

Firstly, relationality as sustainable ethical praxis requires recognition that the relationship is required. Secondly, respect is critical between the parties. Thirdly, each of the parties in the relationship must have equitable rights. Fourthly, for the relationship to be sustainable, the parties must take responsibility for its wellbeing. This involves reciprocity: the giving and the taking, understanding that it is not a one-way street. Finally, when things go wrong, restitution must be made for the damages that have been done.

While most would recognise the truth of this web, relative to the human-to-human context, when it comes to human–other-than-human relationality, such an ethical praxis seems outside the scope of most of contemporary society. Since settler-colonialism spread across at least three continents, this relational practice across the human and other-than-human Indigene has been absent outside of continuing Indigenous cultures (Wolfe 2006). Weaving an Indigenous philosophical animism into the current social context opens a way for an ethical praxis to evolve, one that can bring to the surface the complex relationalities for the purpose of connecting to care and to belong: to the Presences, places, and people.

## 7. Weaving Conclusions for Futures

This paper has explored the place of auto/biography, within the Presences of personal Ancestral storying in the context of Dharug Ngurra, to better understand the relationality that underpins sustainable futures. Drawing out what sustainable relationalities look like and how they have been produced across 65,000 years, requires an ongoing focus on what Indigenous law/lore for living involves. Understanding that the significance of any auto/biographical journey involves the performative (oral, written, visual, audial, kinaesthetic) which expresses lived relationality is critical. Recognising the place of kinship-sensing within that journey is, therefore, also imperative. Weaving the forms, the metaphysical and physical presences, places, and people, across past, present, and future times recognises the possibility for personalised and localised agency that ultimately underpins sustainability. Doing so through the poetic opens the opportunity for storying beyond the 'evidence-base', so that a more inclusive understanding of relationality can be met, one that entwines other-than-humans and our relational agencies.

Showing how relationalities involve the various threads of connection ultimately requires *not* focussing on the disconnections. In this case, the relationalities relevant to this author's family history have not focussed on the unspoken, the hidden contextualities that drove pretence for social 'respectability'. Just as storying can be transgenerational, so can the silencing. In families where such silencing was the tool used for survival, for example, where Dharug language was only used indoors because you could be jailed if caught speaking it publicly, or where grandmother's dark skin colour was narrated as Mediterranean rather than truthfully explained, or where even convict heritage was considered shameful, let alone Aboriginality, so, in these families, the need for survival, comfort, and putting a roof over your and your children's heads was positioned as more important. Once the truth-telling begins, however, the conclusion can only be "Who'd have thought?", the point from which the turn towards sustainable futures begins.

Yanamawa budyari gumada,

Walking with good spirit,

Didgerigur, thank you.

**Funding:** This research received has received external funding from: State of NSW Department of Planning and Environment, Cultural Fire Management Team: 'Dharug women's Cultural Burn' Project Pure Award ID: 199160315.

**Informed Consent Statement:** Informed consent was obtained from all subjects involved in the study.

**Data Availability Statement:** Not applicable.

**Conflicts of Interest:** The author declares no conflict of interest.

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
