# Peer review of "“Who’d Have Thought?”: Unravelling Ancestors’ Hidden Histories and Their Impact on Dharug Ngurra Presences, Places and People"

_genealogy, doi:10.3390/genealogy7020041_

Round 1

Reviewer 1 Report

This is an interesting paper with a great deal of potential.  I think there are some issues that need to be corrected, however,

Some of these are with writing.  There are many incomplete and/or run-on sentences.  

In addition, the writing needs greater clarity.  For instance, in the first three pages there are numerous terms and ideas that need explanation.  What is "auto/biography"?  Is this different from autobiography and biography?  Or does this refer to them both?  Is "Ancestral" different from "ancestral"?  What warrants the capitalization?  Who are the Dharug Ngurra and why is the term sometimes grouped with / Country?

There is a lot that needs to be "unpacked" in these pages.

Terms that might be unfamiliar to readers need to be explained and/or defined when first used.  In addition, when familiar terms are used in unfamiliar ways, this should be explained.  Otherwise, the reader wonders if this is a mistake and, if it is intentional, why is it done this way?

54:  What is "biodiverse justice in a climate changing world."

55:  What is "poetic multimedia"?

58:  What is the "Black Dolly"?

From about line 77, the focus improves, and the reader feels on surer ground.  Up to this point, I don't have a very clear understanding of what the author wishes to convey in the abstract and the introductory section.

Line 150:  It might be worth discussing oral history techniques here -- possible pitfalls as well as advantages with this methodology?

Lines 256-264:  This phenomenon occurs in just about any society in which Europeans, native peoples, and peoples of African descent have mixed.  It is very common in North America (i.e., legends of "Spanish" or "French" or "Italian" ancestry versus "native" or "African" or "mulatto").

272-285:  References to "Dolly" -- since this is mentioned earlier, there needs to be more initial discussion, ideally with references when the term is first used.

316-318 and earlier:  It might be worth contextualizing Pybus more fully.  Her work is very important in Atlantic World scholarship, and her larger project is important for the message of this essay beyond the research on specific individuals discussed here.  You might also reference her Epic Journeys of Freedom study as well as her Black Loyalist website resources online.  

As a general comment, I do not think the references, including the hyperlinks, are in the journal's preferred format.  The long hyperlinks (379-386) might go better in a footnote or in the bibliography than in the text itself.

419:  Define BNI earlier in the paper.

614-616:  Is this the correct placement for this reference?

General notes:  This is a fascinating study, but, especially in the first few pages (Lines 1-80), I think the writing can be sharpened to better frame the article.  There are many terms that a reader unfamiliar with this specific group will not recognize.  In addition, the use of certain familiar terms (auto / biography and/or "Ancestral") strikes me as unusual.  If this is intended, it should be better explained.  After about line 80, the tone becomes more academic, and there is greater contextualization.

Overall, the argument could be strengthened.  More precision with the terminology earlier in the article will help with that.   The concluding section (583-606) could also be sharpened a bit, but I find it more effective than the introduction.  

Overall, the writing is reasonably clear, especially after line 80.  But there are run-on sentences and some questionable word choices.  The writing needs to be reviewed carefully.  Terms need to be defined.  Readers unfamiliar with Australian history may need additional context, and the methodology needs to be clarified and better explained early on.

Author Response

Thank you for your review of this work. I have addressed your concerns through an extended change to the beginning of the paper, responding to your questions around explanation of terms such as auto/biography, biodiverse justice, poetic multimedia, Black Dolly, capitalisation of terms, e.g. Ancestors, positioning these explanations at the beginning, including defining Dharug Ngurra. Referencing questions have also been addressed. 

Reviewer 2 Report

The strength of this paper is in its originality. Not many auto/biographies of racial minorities find their way into publication. This makes such journal articles rare. What this paper does then, is that it gives readers a glimpse into the many racial fabrics woven together over the centuries to make a family and a people. It is a story, a journey of self-discovery and revelation at a period when miscegenation was forbidden but still secretly engaged in. Through the eye of the author(s), we see some level of "truth-telling" that explains how people are related within the long history of a family, and how these people forge(d) a particular relationship with their environment ("other-than-human"). 

The paper's contribution to scholarship is immense because of the rare nature and style of the auto/biography. Among other things, the paper shows us an innovative way to view and approach auto/biography from the prism of a newly-uncovered family history. It is the revelation that the particular auto/biography of the family is therapeutic since it "opens opportunities to heal, decolonise and transform Dharug and, more broadly, Indigenous communities, their knowledges, practices and ontologies." This is an important contribution to scholarship in the form of philosophy and epistemology rooted in ontology and praxis. The author's approach links the very distant past (the ancestral beginnings) to the present and the unfolding future (that of climate change).

While the story the paper tells is meaningful, the quality of structure and clarity are somehow compromised, especially at the beginning. The first two pages are particularly tedious and poorly structured. The abstract (lines 5-20) is repeated verbatim on the second page (lines 40-56). Those latter lines (40-56) should be deleted. The author should also control the verbiage, not only at the abstract but throughout the paper. Some of the sentences are strung together and too long. That style compromises both the quality of the work and the clarity of the message. The author should break up some of the lengthy sentences into smaller ones to allow the meaning(s) emerge to make a strong impact. That could also help control some of the punctuation problems. There are some incomplete sentences that should be taken care of. Book titles should be italicized (see line 169). I think it will also make the paper easier to read and understand if the author settles down to using the word "storytelling" which is among the keywords selected instead of "storying." If the fear is that "storytelling" may make the history being recounted seem like fable, the author could substitute "narrating" for it. Similarly, "narrative" might be a good substitute for "history" instead of bringing Summers' "His-story" and "Her-story." History does not originate from "male dominance" but from the Latin word "historia." I suggest that lines 608-610 be deleted. While they may be sentimental and culturally polite in the Indigenous community, they disrupt the academic rigor, and trivialize the quality, of the paper. Some entries in the "References" also need attention (see lines 614, 642, 643, 697,698). The author should slowly and carefully proofread the entire work to improve the academic quality. 

When the weaknesses pointed out above are corrected, the paper could be considered academically sound. The discussion generated in the paper--that in an auto/biography and/or genealogy of Indigenous peoples such as his family and the Dharug community, it is "safe" to use "oral stories of their Ancestors" instead of written stories as the only evidence of "truth"--is an important one. The paper reveals that "truth" resides in both. "[W]hen it comes to reliance on early colonial-settler identifications, transgenerational oral biographical information is no less reliable than the written records that are available, because their scarcity makes the gaps louder than the 'facts'. While written records may help if, and when, they are available, their absence should not be considered as the legitimate measure for identity verification, when it comes to the earliest settler-colonial invasion of Dharug and Dharawal countries." These are powerful statements that one could agree with. They are a definition of what truth is wherever and whenever it is found--the conformity of the intellect with reality. Such oral histories are not a figment of one's imagination; they are not the rabid conspiracy theories of the present. They are real in the making and tracking of a people.

The sources engaged in the paper are good and recent.       

I have already addressed this above.  The writing must be improved. The author should revise the paper as suggested above. 

Author Response

Thank you for your predominantly positive response to my paper. I have now made significant changes to the introductory section of the work, addressed issues around the length and complexity of sentences, determined to stick with 'storying' as the main term, and adjusted the use of 'storytelling' - there are now only two occurrences of the latter term, which are contextually relevant. Just to respond to your comment on 'historia', while the concept is Latin, the practice of establishing 'history' in the written world has been dominated by the male historians voice, just as in social narratives, and it is therefore appropriate it has been and continues to be contested, so that other voices, such as those of Indigenous and diverse genders also can be heard and seen. Once again, thank you for your positive responses, and considered perspectives.

Round 2

Reviewer 1 Report

The author has done a good job addressing the concerns I raised in my initial review.  In my view, the changes made frame the paper better and more clearly identify the major issues at the outset.  In addition, unfamiliar concepts and terms are better explained in this version, and those terms are addressed and explained earlier in the paper.  The changes to writing addressed the issues I mentioned in my original report, as have done the changes to the references.  The internal editorial team may have additional suggestions concerning writing and references, but these changes address all of my own concerns.

I will say that this is a fascinating study.  The way the author interweaves their own ancestral story with the larger historiographical themes and concepts is most interesting.  Much of the paper's originality lies in this, and the author does a good job showing how individual, particular cases relate to and nuance not only current historical debates but also political and cultural ones.

Author Response

Thank you for your positive opinion of my adjusted work. There is nothing in your response however that suggests that I change anything further so  it is not clear to me why you cannot sign off on it. Or what conclusions prevent you from answering yes:

Are the conclusions thoroughly supported by the results presented in the article or referenced in secondary literature? ( ) (x) ( ) ( )

Accordingly I don't feel I can make any further adjustments at this point.

best wishes,

Reviewer 2 Report

This is a much improved version of the paper. In the References, author should move the first entry, "Histories..."  and place it after the entry, "Hawkins, Ralph..." to conform to alphabetical order. 

Author Response

Thank you for your positive feedback. I have now re-positioned the reference as directed.

best wishes...
